# Oncogenic Properties of the EBV ZEBRA Protein

**DOI:** 10.3390/cancers12061479

**Published:** 2020-06-05

**Authors:** Diego Germini, Fatimata Bintou Sall, Anna Shmakova, Joëlle Wiels, Svetlana Dokudovskaya, Emmanuel Drouet, Yegor Vassetzky

**Affiliations:** 1CNRS UMR9018, Université Paris-Saclay, Institut Gustave Roussy, 94805 Villejuif, France; germinidiego@gmail.com (D.G.); fabisall3007@gmail.com (F.B.S.); anyashm@gmail.com (A.S.); wiels@igr.fr (J.W.); svetlana.dokudovskaya@gustaveroussy.fr (S.D.); 2Laboratory of Hematology, Aristide Le Dantec Hospital, Cheikh Anta Diop University, Dakar 12900, Senegal; 3CIBB-IBS UMR 5075 Université Grenoble Alpes, 38044 Grenoble, France; Emmanuel.Drouet@ibs.fr; 4Koltzov Institute of Developmental Biology, 117334 Moscow, Russia

**Keywords:** EBV, ZEBRA, Zta, BZLF1, lytic cycle, viral-host interaction, oncogenesis, transcription, transactivation

## Abstract

Epstein Barr Virus (EBV) is one of the most common human herpesviruses. After primary infection, it can persist in the host throughout their lifetime in a latent form, from which it can reactivate following specific stimuli. EBV reactivation is triggered by transcriptional transactivator proteins ZEBRA (also known as Z, EB-1, Zta or BZLF1) and RTA (also known as BRLF1). Here we discuss the structural and functional features of ZEBRA, its role in oncogenesis and its possible implication as a prognostic or diagnostic marker. Modulation of host gene expression by ZEBRA can deregulate the immune surveillance, allow the immune escape, and favor tumor progression. It also interacts with host proteins, thereby modifying their functions. ZEBRA is released into the bloodstream by infected cells and can potentially penetrate any cell through its cell-penetrating domain; therefore, it can also change the fate of non-infected cells. The features of ZEBRA described in this review outline its importance in EBV-related malignancies.

## 1. Introduction

Epstein-Barr Virus (EBV) or Human Herpesvirus 4, is a γ-herpesvirus that predominantly infects B-lymphocytes, and, to a lesser extent, epithelial, T and Natural Killers (NK) cells [1,2]. Discovered in 1964 as the first human oncogenic virus, EBV is one of the most widespread human viruses, affecting nearly 90% of the world’s population [3,4]. After an initial infection, it persists as an episome in B cells for the rest of the host’s life. In the vast majority of cases, EBV infection is asymptomatic, but in some individuals, it causes infectious mononucleosis. Furthermore, EBV is associated with various types of malignancies, including Burkitt lymphoma (BL), Hodgkin lymphomas (HL), nasopharyngeal and gastric carcinoma, and post-transplant lymphoproliferative disorder (PTLD). Indeed, EBV was classified as a class I carcinogen by World Health Organization (WHO) [5] and the evidence for EBV-associated oncogenesis have been recently reviewed and updated [6]. The pathogenic and oncogenic effects of EBV are mediated by several EBV proteins and non-coding RNAs. Here we will describe the role of one of EBV immediate early (IE) proteins, ZEBRA, in the viral life cycle and in regulation of the host genome and the consequences thereof.

### 1.1. EBV Life Cycle

EBV life cycle involves a latency phase and a lytic cycle, each associated with specific gene transcription and translation programs. These phases are summarized in Figure 1.

#### 1.1.1. Latency

Following contamination via saliva, primary lytic infection occurs in the epithelial cells of the oropharynx [7] through virus binding to αvβ integrins and the ephrin A2 receptor [8,9]. EBV also infects naïve B cells by interaction with complement receptors 1 and 2 (CR1/CD35 and CR2/CD21) as well as major histocompatibility complex (MHC) class II as a co-receptor [10,11]; this leads to latency establishment and lifelong EBV persistence. Latency can be divided into three successive programs [7,12,13].

The first, called the “growth program” or latency III is characterized by the expression of nine latency proteins (six nuclear proteins: Epstein Barr Nuclear Antigen (EBNA) 1, 2, 3A, 3B, 3C and LP, and three membrane proteins: Latent Membran Protein (LMP) 1, 2A and 2B); two Epstein Barr virus-encoded small non-coding RNAs (EBER1 and 2), BHRF1 miRNA and BamHI-A fragment rightward transcript (BART) transcripts [14,15]. EBV-infected B cells are activated and differentiate into proliferating B blasts. This phase triggers a powerful immune T cell cytotoxic response; however, it is usually insufficient to eliminate all infected B cells [12].

Remaining B blasts migrate to the tonsil germinal center (GC) where the second program, called the “default program” or latency II, occurs. Infected cells turn off the expression of all latency proteins except EBNA1 and LMP1 and 2. These proteins are thought to promote B blasts differentiation into centroblasts and then centrocytes [16].

The last program is named the “latency program” or latency 0. Centrocytes differentiate into resting memory B cells which leave the GC and circulate in peripheral blood without any latency protein expression [17]. Finally, during latency I, cell division of infected memory B cells occurs, the weakly immunogenic EBNA1 is expressed to ensure latent viral genome replication and its persistence within daughter cells [18].

#### 1.1.2. Lytic Cycle

Occasionally, following stimulation, infected memory B cells may be recruited into GC and then either reintegrate the memory cell reservoir or differentiate into plasma cells and reactivate the EBV lytic cycle. In healthy carriers, lytic reactivation is observed only in plasma cells [19]. It is characterized by sequential expression of lytic cycle proteins resulting in production of new infectious viruses and lysis of infected cells [7,13].

Studies of the defective EBV genome helped to identify a transcription factor encoded by the EBV *BZLF1* gene as the key actor in switching from latency to lytic phase [20]. This protein, named ZEBRA, Zta, Z, BZLF1 or EB-1, when expressed in latently infected cells, activates the entire EBV lytic cycle cascade [21]. ZEBRA also activates transcription of the second IE gene *BRLF1* coding for the RTA transcription factor. ZEBRA and RTA function synergistically to activate the early genes involved in metabolism and viral DNA replication and the late genes encoding for EBV structural proteins [4].

Thus, EBV has two tightly regulated latent and lytic phases characterized by specific gene expression patterns. However, there is evidence that both latent and lytic gene expression may be simultaneously present within the same cell. *BZLF1* expression in freshly infected B cells starts as early as 1.5 h post-infection and lasts for several days. In these cells, transcription of the late gene *BLLF1* was not detected suggesting a partial activation of the lytic cycle [22]. This stage, characterized by IE and early gene expression without production of new virions or cell lysis, is commonly referred to as an abortive lytic cycle [23,24] or transient pre-latent abortive lytic cycle when it occurs immediately after infection [25]. Only a minority of EBV-infected B lymphocytes from healthy carriers completes the lytic cycle after stimulation, the vast majority generating an abortive lytic cycle [26]. However, how this abortive lytic cycle takes place in vivo remains unclear.

### 1.2. EBV-Related Oncogenesis

Despite its asymptomatic persistence in most of the adult population worldwide, in a minority of individuals, EBV is strongly associated with several non-malignant diseases such as infectious mononucleosis, chronic active infection, hemophagocytic lymphohistiocytosis, oral hairy leukoplakia and autoimmune diseases [2,27]. The vast majority of EBV-associated diseases are however represented by cancers occurring both in immunocompetent hosts (Table 1) and in patients with primary or acquired immunodeficiency (Table 2). They are mostly B cell malignancies (BL, HL, PTLD, diffuse large B cell lymphoma (DLBCL)), nasopharyngeal carcinoma (NPC) or, less frequently, T cell malignancies, gastric, breast and hepatocellular carcinomas, leiomyosarcoma and follicular dendritic sarcoma [1,2,28]. Many mechanisms of EBV related oncogenesis have been proposed and a possible role for different EBV components has been described (reviewed in [7,27,29,30,31,32]). Nevertheless, even if great progress has been made in understanding the EBV links to cancers, many aspects of EBV-related oncogenesis are still unknown and represent a major challenge in cancer research.

EBV genome contains approximately a hundred genes coding for latency and lytic cycle proteins and many small non-coding RNAs expressed to ensure the normal life cycle of the virus. Expression of some proteins and RNAs have been correlated with development of EBV-associated malignancies. The oncogenic properties of each EBV latent protein has been extensively explored (reviewed in [1,2,7]); however the involvement of EBV lytic cycle in oncogenesis is no less important than the contribution of individual proteins. Even though the lytic cycle was long assumed to inhibit tumorigenesis due to final lysis of the infected cells, an increasing amount of data support its contribution to oncogenesis mainly at its initiation or through the abortive lytic cycle and/or autocrine or paracrine effects of EBV IE proteins [27,34,35]. ZEBRA could thus be seen as an important player in EBV-driven oncogenesis, in collaboration with other viral and cellular proteins since it induces the lytic cycle. Additionally, ZEBRA can exit EBV-infected cells either by secretion or after cell lysis and potentially penetrate other cells through its cell penetration domain (CPD) [36]. In EBV-infected cells, ZEBRA alone can switch EBV from latency to lytic cycle; therefore, it may transduce reactivation signals between infected cells. ZEBRA can also reactivate transcriptionally silent host genes due to its affinity to methylated promoters [37] and thus affect key cellular pathways implicated in oncogenesis, control of cell cycle, proliferation and apoptosis.

We will next discuss the structural and biological properties of ZEBRA to provide insights into its potential oncogenic activity and clinical applications.

## 2. ZEBRA Structure and Functions

### 2.1. ZEBRA Domain Organisation

ZEBRA is encoded by the EBV *BZLF1* gene, transcribed to a mRNA composed of three exons and translated into a 27 kDa protein containing 245 amino acids (Figure 2A).

ZEBRA belongs to the family of basic leucine zipper (bZIP) transcription factors. Its bZIP domain (residues 175–221) consists of the central basic DNA binding domain (DBD, residues 178–194) and the C-terminal coiled-coil dimerization domain (DD, residues 195–221) [38,39]. ZEBRA homodimer grasps DNA via its two long helices, with the DBD contacting the major groove and DD forming a coiled coil. A185 and S186 of ZEBRA directly interact with methylated cytosines in DNA [37].

Unlike eukaryotic bZIP factors, ZEBRA lacks a classical heptad repeat of the leucine zipper motif [40], but its bZIP domain is additionally stabilized by the C-terminal tail, which makes a turn and runs antiparallel to the coiled coil [39]. Residues 167–177 are considered as the “regulatory domain” and their phosphorylation can modulate ZEBRA activity [38,41].

In the N-terminal transactivation domain (TAD, residues 1–166), the residues 52–64 and 77–86 are rather unstructured and amorphic [42,43] (Figure 2A). The basic region within TAD (residues 157–161) is considered to be important for recognition and high affinity binding to methylated DNA [44].

ZEBRA can exit and enter cells and nuclei due to its CPD situated within the bZIP domain between residues 170 and 220 [36]. This CPD was successfully used to transduce human cells in vitro [36,45]. CPDs are short sequences with a composition that enables them (and the adjacent protein) to penetrate cells either via endocytotic entry followed by endosomal escape, or by directly penetrating the cell membrane. Their composition is usually either cationic (with a high number of positively charged residues) or amphipathic (with hydrophilic and hydrophobic regions of residues) [46]. ZEBRA CPD is rich in positively charged residues (seven lysines and seven arginines), mostly within DBD (basic region) (Appendix A in blue), whereas hydrophobic amino acids (one valine, five alanines, seven leucines) of CPD are mostly within DD (leucine zipper) (Appendix A in red) [36]. In another protein possessing the CPD, human immunodeficiency virus (HIV)-1 Tat, the CPD region is also multifunctional [47]. Presumably, the cationic part serves for interaction with the negatively charged phosphate groups of membrane phospholipids as well as on DNA, while hydrophobic residues interact with the hydrophobic part of the phospholipid membrane and participate in ZEBRA’s dimerization (Appendix A).

When entering cells, ZEBRA is targeted to the nucleus and has a pan-nuclear localization, with the exception of the nucleolus [48]. Substitution of several amino acids within DBD can alter subnuclear localization from pan-nuclear to focal [49]. The bipartite nuclear localization sequence of ZEBRA is located within DBD (residues 178-194), but a small region within TAD (residues 157-162) is also important for the nuclear import [50].

To summarize, ZEBRA structure accounts for its important functions because the chromatin-binding capacity via its DBD and DD and its ability to act as a transcriptional activator thanks to its TAD allows it to regulate expression of both viral and host genes [36,51]. ZEBRA also possesses a CPD that allows it to penetrate into uninfected cells [36].

### 2.2. Posttranslational Modifications of ZEBRA

ZEBRA is prone to posttranslational modifications. It is constitutively phosphorylated in vivo at multiple sites mostly clustered within TAD and the regulatory domain [52]. Phosphorylation of S173 and to a lesser extent S167 within the regulatory domain is important for DNA binding [41,53]. Constitutive phosphorylation may also explain why ZEBRA’s apparent mass on gel electrophoresis is 35 kDa [40] instead of the predicted 27 kDa [54].

ZEBRA also has a sustained N-terminal M1 acetylation [55]. K12 of ZEBRA is a substrate of partial and reversible SUMOylation [56,57] that affects neither protein stability nor its subcellular localization but significantly decreases ZEBRA transactivation activity by inhibiting its binding to CBP (CREB-binding protein) (see below) [57,58]. SUMOylation is diminished in DD-deficient ZEBRA [57]; EBV-encoded protein kinase also reduces ZEBRA’s SUMOylation, and this effect is not related to S209 phosphorylation, conventional site of ZEBRA modification by this kinase [59].

Thus, ZEBRA is extensively and mostly constitutively modified after translation, presumably by certain viral and host enzymes. The regulation of posttranslational modifications, their role and possible regulatory potential for ZEBRA activity and the EBV status remain to be elucidated.

### 2.3. ZEBRA Functioning in Host Cells

ZEBRA functions in host cells rely on its capacity to bind specific DNA motifs and interact with other proteins. DBD of ZEBRA binds to heptamer DNA motifs, named ZEBRA response elements (ZREs). ZREs are present within both viral and host gene promoters. At present, two types of ZREs are identified: an activator protein 1 (AP-1)-like recognition elements (non-CpG-containing) [40] and CpG-containing recognition elements [60] (Figure 2B). Binding to CpG-ZREs depends on DNA methylation [60,61]. During latency, EBV genome becomes heavily methylated to suppress its transcription, however, ZEBRA binds methylated promoters with high affinity and activates gene transcription to initiate lytic cascade [44]. ZEBRA’s selectivity and preference for methylated DNA is a key to hijacking host epigenetic silencing, which is important for EBV latency reversal, oncogenesis, control of cell cycle, proliferation and apoptosis [37]. Ten-eleven translocation methylcytosine dioxygenase that reduces ZEBRA binding to methylated promoters [37] can be considered as ZEBRA host restriction factor.

#### 2.3.1. Transcriptional Regulation

ZEBRA can both activate and downregulate transcription of viral and host genes. Transcriptomic analysis of B cells with ectopic expression of ZEBRA revealed 2263 deregulated genes (74% upregulated, 26% downregulated) [49]. Upregulated genes include those involved in cell adhesion, morphogenesis, projection and response to hormones, while downregulated genes are involved in the immune response, induction of apoptosis and lymphocyte activation [49]. In total, 12% of these genes (207 upregulated and 71 downregulated) are directly regulated by ZEBRA which binds to their promoters, as identified by chromatin immunoprecipitation followed by sequencing (ChIP-seq).

During activation of lytic cycle, ZEBRA binds promoters of early lytic viral genes and host genes and, via its TAD, interacts with basal transcription factors IID [63] and IIA [64] (TFIID and TFIIA); this leads to sequential recruitment of other basal transcription factors and RNA polymerase II (Figure 3A). In addition, ZEBRA binds the transcriptional coactivator and histone acetyltransferase CBP (CREB binding protein) which increases ZEBRA transactivation properties [65]. Direct binding to the Transducer of Regulated CREB coactivator enhances ZEBRA-mediated transcription [66].

Transcriptional repression by ZEBRA is related to its specific binding to cellular transcription factors mainly via its bZIP or TAD. In most cases, such interaction mutually impedes their function as transcription factors and results in repression of target genes for both ZEBRA and the associated transcription factors [67,68].

Direct binding to p53 [69,70], p65 [71] and c/EBP family of transcription factors [68] inhibits their transcriptional activity. ZEBRA directly binds B cell specific transcription factors Pax5 and Oct2 via bZIP domain; this inhibits ZEBRA activity, however, the reciprocal inhibition was proven only for Pax5 [72,73]. Unidirectional inactivation of the family of nuclear factor of activated T cells (NFAT) transcription factors, involved in calcium signal transduction, by direct interaction with ZEBRA was also reported [74]. Presumably, the same mechanism related to ZEBRA inhibitory binding to host transcriptional factor is involved in the class II transactivator (CIITA) repression, however, it involves the TAD and a transcriptional factor inhibited by ZEBRA was not identified [67]. ZEBRA also binds to interferon regulatory factor 7 (IRF-7) through its TAD, decreasing the transcription of interferon (*IFN*) *α4*, *IFNβ,* and antigen presentation 2 (*Tap-2*) [75]. Finally, SUMOylation of ZEBRA appears to be important for transcriptional repression since it promotes recruitment of histone deacetylases to responsive promoters [57] (Figure 3B).

#### 2.3.2. Binding to the Replication Origin in EBV Lytic Replication

During the lytic cycle, ZEBRA binds EBV lytic origin (oriLyt) and recruits viral core replication enzymes to initiate replication [76,77] (Figure 3C). In contrast to latent replication, EBV lytic replication relies on virally encoded replication enzymes, whose expression is induced during the lytic cycle: helicase (*BBLF4*), primase (*BSLF1*), primase-associated factor (*BBLF2/3*), DNA polymerase (*BALF5*), DNA polymerase processivity factor (*BMRF1*), and single-stranded DNA binding protein (*BALF2*) [41]. This function is mediated by the TAD (residues 11–25), which interacts with the viral helicase, primase and DNA polymerase [78,79]; and by the bZIP domain which interacts with the DNA polymerase processivity factor [41]. S173 phosphorylation within the regulatory domain is essential for ZEBRA action as a replication factor [41].

#### 2.3.3. Interaction with Other Cellular Proteins

ZEBRA also interacts with proteins other than transcription factors (Figure 3D). For example, ZEBRA interaction with Cul2 and Cul5 induces the formation of the multimolecular ECS complex (Elongin B/C-Cul2/5-SOCS-box protein) that ubiquitinates p53 for proteasomal degradation [80].

Other ZEBRA cellular partners include mitochondrial single-stranded DNA binding protein (mtSSB) [81], nuclear protein 53BP1, a component of the ATM DNA damage response pathway [82], INO80 chromatin remodeler ATPase [83]; these interactions are important for EBV lytic cycle reactivation and replication.

In summary, ZEBRA binds specific DNA motifs and/or interacts with other proteins, either recruiting them to DNA binding sites or altering their activity. However, ZEBRA direct interactions with many other cellular proteins [74] are much less studied as compared to interaction with chromatin-binding proteins.

## 3. EBV-Related Diseases and Oncogenic Properties of ZEBRA

### 3.1. ZEBRA Implication in EBV-Related Malignancies

Increasing evidence supports that *BZLF1* gene expression could contribute, directly or indirectly, to EBV-induced tumorigenesis. ZEBRA protein and mRNA were detected in more than 80% of biopsies from 44 PTLD patients [84]. Lymphoblastoid cell lines (LCLs) derived with wild-type (WT) EBV are more prone to induce a lymphoproliferative disorder when injected into Severe Combined Immunodeficient (SCID) mice than LCLs derived with *BZLF1*-KO EBV [85]. Interestingly, the same results were observed after acyclovir treatment, which inhibits viral DNA replication but not *BZLF1* expression. These data suggest that ZEBRA, and not the production of infectious viral particles, is required for tumor formation in SCID mice [85]. These results were also confirmed in a humanized mouse model where both human fetal CD34+ hematopoietic stem cells and human thymus/liver tissues were transplanted. Indeed, in this model, the development of CD20+ DLBCL was more frequent in mice infected with WT EBV as compared to *BZLF1*-KO EBV [86]. Soluble ZEBRA can be detected in the serum of PTLD patients at concentrations up to 4 µg/mL and it is significantly higher in PTLD patients compared to transplanted patients without PTLD [87]. ZEBRA is also present in serum samples from immunocompromised humanized mice developing lymphoma, with a correlation with tumor mass [35]. The presence of ZEBRA protein or its mRNA was also reported in tumor cells or in tumor tissue biopsies in other types of EBV-induced lymphomas, such as HL, DLBCL and BL [88,89,90,91].

In some EBV-associated lymphoma, there is also evidence for indirect action of ZEBRA. *BHRF1* and *BALF1*, two EBV early lytic genes whose expression is induced by ZEBRA, are found highly expressed in DLBCL [89]. The products of these two genes are the viral Bcl-2 homologs required for B cells immortalization [92]. Moreover, EBV cofactors for endemic BL (*Plasmodium falciparum*, *Euphorbia tirucalli* and potentially Aflatoxin B1) are all able to reactivate EBV in vitro and in vivo [93,94,95].

High ZEBRA expression at mRNA or protein level was also reported in NPC biopsies [96,97] and in breast carcinoma [98]. High anti-ZEBRA IgG titers in sera correlate with poorer clinical outcome in patients [99,100]. The presence of anti-ZEBRA antibody has a high diagnostic accuracy for early-stage NPC [101]. More generally, EBV replication and expression of some early lytic cycle genes were detected in EBV-induced epithelial malignancies including NPC [102,103]. In addition, EBV-infected individuals with elevated titers of IgA antibodies against EBV lytic viral capsid antigen (VCA) have a higher risk of NPC [104]. A subset of EBV-associated gastric carcinoma and some NPC cells also express early lytic genes such as *BHRF1*, *BALF1*, *BARF1* and *BGLF5* [105,106,107,108]. A specific EBV strain isolated from NPC and gastric carcinoma has an enhanced capacity for spontaneous lytic replication and therefore ZEBRA expression [109,110].

### 3.2. ZEBRA Oncogenic Properties

In this section, we will discuss the mechanism by which ZEBRA contributes to acquisition of cancer hallmarks by cells (Figure 4).

#### 3.2.1. Genome Instability

Genome instability (GI), one of the major factors of oncogenic transformation, may result in random mutations and chromosomal rearrangements which can confer selective advantage to certain cells through oncogene activation, downregulation or loss of tumor suppressor genes [111]. GI can occur through different mechanisms: (1) DNA damage production with incapacity to detect damaged DNA; (2) DNA damage with defects in DNA repair; (3) defects in preventing the action of potential mutagens [112,113,114].

Although ZEBRA can interact with proteins implicated in DNA damage response (e.g., 53BP1, a component of the ATM pathway) [82], not much data exist in support of a direct relationship between GI and ZEBRA. However, some events such as oxidative stress that lead to GI occur following EBV reactivation and can thus be related to ZEBRA expression [115,116]. Oxidative stress was also described in purified B cells and epithelial cells at an early stage of EBV infection when ZEBRA is expressed [117]. Furthermore, several early lytic proteins induced by ZEBRA may participate in GI. For example, BMRF1 induces centrosome amplification and chromosome instability in B cells in vitro and in vivo in a mouse model [118]. BGLF4 directly or indirectly induces DNA damage by retarding cellular S-phase progression or inducing premature chromosome condensation associated with a high risk of chromosomal breaks at common fragile sites [119,120,121]. EBV DNase was also found to induce GI in human epithelial cells through DNA damage induction and DNA repair repression [122]. BALF3 has also been linked to DNA strand breaks induction, resulting in copy number aberrations accumulation in NPC cells [123]. Recurrent chemical reactivation of EBV in NPC cells appears to induce GI [124]. EBV reactivation in LCLs induces global nuclear architecture remodeling that could enhance formation of chromosomal translocations [125].

#### 3.2.2. Tumor-Promoting Inflammation

Inflammation favors tumor development and progression. This could be related to high levels of cytokines, chemokines and growth factors observed upon EBV reactivation, including interleukin (IL)-8, IL-10, IL-6, IL-13, Transforming Growth Factor-beta (TGF-β) [25]. ZEBRA can also directly transactivate *IL-8* promoter through its two ZREs, resulting in *IL-8* upregulation in NPC cells [126]. ZEBRA expression in NPC is also associated with upregulation of growth related oncogene and macrophage inflammatory protein-1β [126].

ZEBRA can also bind directly to the human IL-10 (*hIL-10*) minimal promoter to induce transcription of *hIL-10* during the early phase of the lytic cycle in EBV-infected B cells [127]. IL-10 is upregulated in breast cancer and NPC [127,128,129]. A viral analog of the hIL-10 (vIL-10) encoded by the *BCRF1* gene can be produced during the lytic cycle [130].

Other interleukin genes contain ZREs in their promoters [25] and IL-6 and IL-13 production can be directly activated by ZEBRA in infected cells [131,132,133]. ZEBRA also increases expression of the genes coding for *TGF-β* [134] and the Vascular Endothelial Growth Factor (*VEGF*) [135] in B cells undergoing lytic cycle. Finally, ZEBRA can also induce inflammatory cytokines through expression of the early lytic gene *BLLF3,* which can activate NF-κB and induce secretion of pro-inflammatory cytokines (tumor necrosis factor (TNF)-α, IL-1β, IL-6, IL-8 and IL-10) in human monocyte-derived macrophages [136,137].

#### 3.2.3. Immune Evasion

ZEBRA-induced viral and human IL-10 production protects infected cells from immune recognition and elimination. Indeed, IL-10 interferes with antiviral cytokines and NK/NKT cell-mediated lysis [130]. Moreover, IL-10 downregulates transporter proteins associated with TAP1 and consequently induces a reduction of surface MHC I molecules on infected B lymphocytes [138]. IL-10 also inhibits IFNγ release which plays a central role in resistance of the host to infection [139].

ZEBRA also promotes immune evasion by disrupting cell signaling pathways activated by IFNγ such as the JAK-STAT pathway. It decreases IFNγ receptor expression and inhibits phosphorylation of Jak1, Jak2 and STAT1 molecules consequently downregulating their downstream target genes, including *MHC II* [140]. ZEBRA can also directly thwart surface expression of MHC II molecules by transcriptional repression of *CIITA*, a main regulator of human leukocyte antigen (HLA) class II genes [141] and *CD74*, the invariant chain of MHC II that facilitates its transport to the cell surface [142].

The immunomodulatory effects can be induced by three EBV lytic cycle proteins called immunoevasins (*BILF1*, *BNLF2a* and *BGLF5*). They interfere with host antigen processing pathways and consequently allow EBV-infected cells to escape from immune system action. BILF1 reduces MHC I molecules on the cell surface by physical interaction and inhibits CD8+ T cell recognition of endogenous target antigens [143,144]. BNLF2, a TAP inhibitor, impairs peptide loading onto HLA class I molecules thus blocking antigen presentation to cytotoxic T cells [130,145]. BGLF5, the EBV alkaline exonuclease, downregulates HLA class I and II impairing antigen recognition by immune cells [146].

It is noteworthy that the described immunomodulatory effects induced by ZEBRA are associated not only with cancer progression but also with development of autoimmune diseases, e.g., systemic lupus erythematosus [147].

#### 3.2.4. Cell Proliferation and Growth

ZEBRA-induced IL-10 enhances the viability of resting B lymphocytes and supports growth and differentiation of EBV-infected cells [127,139]. Both IL-6 and IL-13 promote proliferation of EBV-infected cells and long term growth of LCLs [131,133]. In agreement with this, growth of both LCLs and EBV-induced B cells after primary infection can be inhibited by treatment with an anti-IL-13 antibody [133]. Treatment with an anti-IL-6 antibody led to remission of B-lymphoproliferative disorder in eight out of 12 patients studied [148].

ZEBRA-induced IL-8 may be used by some tumor cells as an autocrine growth factor [149]. The early lytic gene *BARF1* also possesses an autocrine mitogenic activity and is an in vivo growth factor [150]. *BARF1* is associated with Cyclin D1 overexpression in EBV-associated gastric cancer [151].

#### 3.2.5. Resistance to Cell Death

The most common anti-apoptotic effect following ZEBRA expression is through activation of the two viral Bcl-2 homologs *BHRF1* and *BALF1*. Another EBV early lytic gene, *BARF1,* can activate the cellular anti-apoptotic protein Bcl-2 in fibroblasts [152]. *BARF1* expression leads to an increased Bcl-2 and Bax ratio and decreased PARP cleavage in gastric carcinoma cells [153].

*BZLF1*-KO LCLs showed a significant increase in the percentage of dead cells, reversible after *BZLF1* expression, whereas no difference was observed between *BRLF1*-KO and WT LCLs, thus suggesting a direct ZEBRA-mediated anti-apoptotic effect [85]. ZEBRA also downregulates the expression of tumor necrosis factor receptor 1 (TNFR1) by direct binding to its promoter [68,154]. This prevents TNF-α activation and consequently TNF-α induced apoptosis.

#### 3.2.6. Other Oncogenic Effects

ZEBRA can positively affect tumor progression by inducing the expression of *VEGF* and *IL-8*, both associated with angiogenesis, tumor development, metastasis and resistance to chemotherapy [135,155,156]. Moreover, expression of ZEBRA by tumor cells from NPC patients correlates with advanced lymph node metastasis, and this effect has been related to direct transactivation of the Matrix Metalloproteinase (*MMP9*) promoter by ZEBRA [157]. In addition to *MMP9*, ZEBRA can also induce *MMP3* upregulation in epithelial cells by binding to the ZRE in the *MMP3* promoter. Both MMP3 and MMP9 act in synergy to promote tumor invasion and metastasis [158]. The oncogenic early lytic gene *BARF1* enables replicative immortality through induced activation of telomerase in primary epithelial cells [159].

ZEBRA, through its bZIP domain, can also directly interact with cancer-associated transcription factors: p53, RAR, CBP, and C/EBPα [160,161,162,163]. These interactions have a functional importance for EBV life cycle, but a direct demonstration for their role in human disease is still lacking even if these ZEBRA targets are often deregulated in cancers [164,165].

Interestingly, ZEBRA has a functional homology with another viral protein: the HIV-1 transactivator protein Tat. Like ZEBRA, Tat can be secreted into the bloodstream by HIV-1 infected T cells and, through its cell penetration domain, can enter uninfected cells, including B cells. Tat protein is potentially oncogenic in B cells since it induces a relocation in the nuclear space of the *MYC* locus close to the *IGH* gene in circulating B-cells [166] and an overexpression of the *AICDA* gene coding for the Activation-induced Deaminase (AID) [167]. These two events promote formation of BL-specific translocations and could at least partly account for the high frequency of BL in HIV-infected patients. Tat also promotes mitochondrial production of reactive oxygen species (ROS) and thus DNA damage and genome instability in B cells [168]. Interestingly, ZEBRA and Tat can be present in a cell at the same time in the blood of HIV-infected individuals and possibly interact. This hypothetical interaction could have an antagonistic or synergistic effect on their oncogenic activity.

## 4. ZEBRA in Diagnosis and Therapy

Many articles report the presence of EBV lytic cycle in tumor cells from HL [169], NPC [99,170,171,172,173,174], in transplant patients [175], and in breast tumors [98]. Clinical studies on EBV lytic proteins including ZEBRA in patients with PTLD or HIV-associated non-Hodgkin lymphoma NHL are mostly related to the role of these proteins in neoplastic tissues [84,89,176,177]. Both high EBV copy number and strong *BZLF1* mRNA expression in the peripheral blood lymphocytes (PBL) of patients are sensitive markers of EBV-related PTLD [178]. ZEBRA was expressed in 5% of whole peripheral blood mononuclear cells from a patient with a lymphoproliferative disease who underwent non-myeloablative allogeneic stem cell transplantation [179]. Moreover, the number of EBV-infected cells in the peripheral blood increases after immunosuppression: on average, 1.6 latently-infected cells per 10^4^ B lymphocytes [180] vs. 12.5 per 10^6^ B cells in persistently infected healthy individuals [181].

Soluble ZEBRA concentrations of >100 ng/mL detected by an enzyme-linked immunosorbent assay (ELISA) in serum of patients after solid organ or hematopoietic stem cell transplant were predictive of PTLD in 80% of the cases within three weeks [87]. Interestingly, the circulating ZEBRA could be detected during periods in which the viral DNA was not detectable by qPCR. For example, in two patients, ZEBRA was detected at 2 and 6 weeks, respectively, prior to the PTLD episode and before an increase in qPCR signals. Thus, ZEBRA testing in serum could help identify patients likely to develop severe outcomes during the critical posttransplant period and serve as a potential diagnostic marker for EBV follow-up in immunocompromised patients.

The relevance of EBV lytic cycle to human pathology prompted researchers to target certain lytic proteins with therapeutic aims. Adenovirus vectors expressing BZLF1 or BRLF1 were used to treat EBV-positive tumors [182]. On the other hand, Food and Drug Administration (FDA)-approved leflunomide, which targets EBV replication, was shown to inhibit the earliest step of lytic EBV reactivation (*BZLF1* and *BMRF1* expression) and prevented the development of EBV-induced lymphomas in both a humanized mouse model and a xenograft model [183]. More recently, duvelisib (a molecule inhibiting the PI3K/AKT signaling pathway, and B cell receptor (BCR) signaling) was shown to reduce cell growth and expression of EBV lytic genes *BZLF1* and *gp350/220* in EBV-positive cell lines [184]. The histone deacetylase (HDAC) and DNA methyltransferase inhibitors are also possible avenues to suppress the ZEBRA expression and the entire lytic cascade [185].

Immunotherapeutic approaches, such as vaccination against IE proteins or IE-specific therapeutic monoclonal antibodies also represent a promising approach. A recent study demonstrated that vaccination of hu-PBL-SCID mice against ZEBRA could enhance specific cellular immunity and significantly delay the development of the lethal EBV-related lymphoproliferative disease [186].

## 5. Conclusions and Remaining Questions

The role of ZEBRA in EBV infection, lytic cycle and oncogenesis has been extensively studied, but numerous questions remain:

**Abortive lytic cycle:** ZEBRA can affect host cells by inducing the abortive lytic cycle in B cells (production of early EBV lytic proteins without cell lysis); however the fate of these cells remains unclear: they may reintegrate the memory cell reservoir after the abortive lytic cycle, return to latency 0 profile or restart the latency cycle as for the primary infection (Figure 1). Another remaining question is whether some stimuli are more prone than others to specifically induce the abortive lytic cycle. *In vitro,* EBV reactivation stimuli such as stress inducing agents, ROS, anticancer drugs or hypoxia [14,187] directly reactivate the virus in EBV-positive cell lines. Thus, in vivo, the abortive lytic cycle may occur after stimulation by these stress-induced agents, instead of an immunological stimulation which mainly leads to a productive lytic cycle in plasma cells.

**A role of ZEBRA in oncogenesis**: ZEBRA upregulates the transcription of host cell genes coding for cytokines involved in inflammation, angiogenesis, metastasis and cell proliferation. ZEBRA downregulates the expression of MHC II class genes thus promoting the immune evasion and genes related to apoptosis thus inducing cell death resistance. ZEBRA also interacts with cancer-related cellular proteins altering their activity. The net cellular effect of these interactions is quite complex and depends on protein localization, concentration, nuclear architecture, nature of promoters involved. Indeed, for p53 signaling pathway, both inhibitory and stimulatory effects of ZEBRA have been described [69,164,165]. This activity of ZEBRA needs additional studies.

**A role of ZEBRA in non-infected cells:** ZEBRA is released into the bloodstream by infected cells and, due to its CPD, can potentially penetrate into uninfected cells and alter their transcriptional program either directly or via interaction with cellular proteins. These potentially oncogenic effects of ZEBRA in non-infected cells are worth investigating and could link EBV to other as yet unidentified pathologies, independently of EBV presence in cells, thus potentially expanding the spectrum of EBV-associated diseases.

**ZEBRA interaction with other proteins:** the cellular interactome of ZEBRA needs further investigation to explain the functional significance of ZEBRA interaction network [74]. ZEBRA is extensively modified in vivo [52], however, the enzymes (viral and cellular) and signaling pathways involved in its post-translational modifications are largely unknown, as well as the effect of these modifications on ZEBRA activity. A splicing variant of ZEBRA was also described but its functional role is poorly understood [188]. Some EBV strains as well as sequence variations in the *BZLF1* gene may have an enhanced ability to reactivate the lytic cycle [109,110,189]. A better characterization of the variations in the structure of the ZEBRA protein produced by these different virus strains could be relevant.

The relevance of the lytic cycle and the role of ZEBRA in lymphomagenesis is a new paradigm pertaining to the prevention and treatment strategies for EBV-associated cancers. It is therefore important to investigate the lytic EBV infection in immunocompromised patients, such as organ transplant recipients, who are highly prone to developing EBV-associated malignancies. More efforts should be invested to examine the potential of drugs that target EBV lytic proteins, including ZEBRA.

**ZEBRA as a biomarker** (mRNA, anti-ZEBRA IgG and soluble ZEBRA concentration in blood) has mainly been studied in PTLD. It would be important to test whether circulating ZEBRA could serve as a biomarker for other EBV-associated diseases, especially those with the lytic cycle involvement, e.g., endemic BL.

A better understanding of the mechanisms underlying ZEBRA activity in cells will shed light on its role in oncogenesis and open perspectives in early diagnosis and treatment of EBV-related cancers.

## Figures and Tables

**Figure 1 cancers-12-01479-f001:**
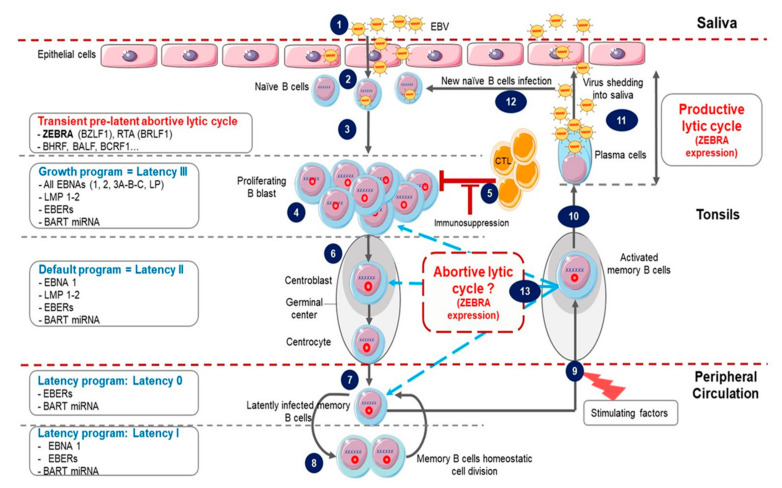
Epstein-Barr Virus (EBV) life cycle. (**1**) Infection occurs after the contact with an infected saliva. (**2**) After initial infection of oropharyngeal epithelial cells, the virus passes into the underlying lymphoid tissue where it infects naive B cells. (**3**) This immediately triggers the transient pre-latent lytic cycle with expression of ZEBRA and other lytic genes involved in resistance to apoptosis and evasion from the immune system. (**4**) Infected naive B cells become proliferating B blasts through the growth program (latency III) where all latency proteins are expressed. (**5**) Cytotoxic T lymphocytes (CTL) trigger a strong immune response (which is impaired during immunodeficiency) to eliminate EBV-infected B cells. (**6**) Proliferating B blasts migrate into the germinal center (GC) and activate the default transcription program (latency II) where latency protein expression is restricted to EBNA1, LMP1 and LMP2. They differentiate into centroblasts and then centrocytes. (**7**) Centrocytes leave the GC and differentiate into memory B cells circulating in peripheral blood. These cells have turned off the expression of all viral proteins (latency 0). (**8**) Occasionally, circulating EBV-positive memory B cells express EBNA1 during homeostatic cell division to ensure viral genome replication and segregation into daughter cells. (**9**) Following stimulation, latently infected memory B cells can be recruited into GC. (**10**) Activated EBV-positive memory B cells can differentiate into plasma cells, reactivate the virus and undergo productive lytic cycle that leads to (**11**) viral shedding into saliva and (**12**) new naive B cells infection. (**13**) Activated EBV-positive memory B cells reintegrate the pool of memory B cells. It is not clear whether in vivo stimulated EBV-positive memory B cells which have not differentiated into plasma cells undergo an abortive lytic cycle (ZEBRA and early gene expression without viral production) before reintegrating the pool of memory cells. It is also not clear whether these cells successively re-express different latency programs in the GC in vivo before reintegrating the pool of memory cells.

**Figure 2 cancers-12-01479-f002:**
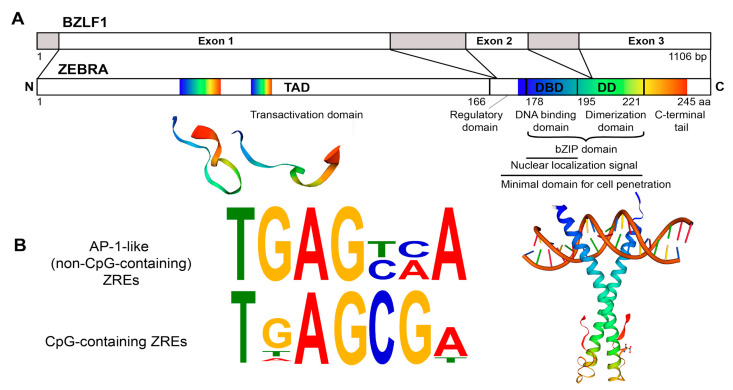
Structure of the ZEBRA protein. (**A**) ZEBRA structure. ZEBRA is encoded by the *BZLF1* gene containing three exons. ZEBRA protein has an N-terminal transactivation domain (TAD, residues 1-166), a regulatory domain (residues 167–177), a bZIP domain, which consists of a central basic DNA binding domain (DBD, residues 178-194) and a C-terminal coiled-coil dimerization domain (DD, residues 195–221). The minimal domain for cell penetration is located between residues 170-220. Three available partial 3D structures were imported from the SWISS-MODEL Repository [62] (accession number P03206) and are based on crystal structure data published by [39,42,43]. They are shown below the respective primary sequence. Rainbow color code is used to map approximate residue position concordance between primary and tertiary (or quaternary) structure. (**B**) ZEBRA-response elements (ZREs). Sequences of ZEBRA DNA binding sites (ZREs) of two types: AP-1-like (non-CpG-containing) ZREs and CpG-containing ZREs are depicted as sequence logos, adapted from [51,60].

**Figure 3 cancers-12-01479-f003:**
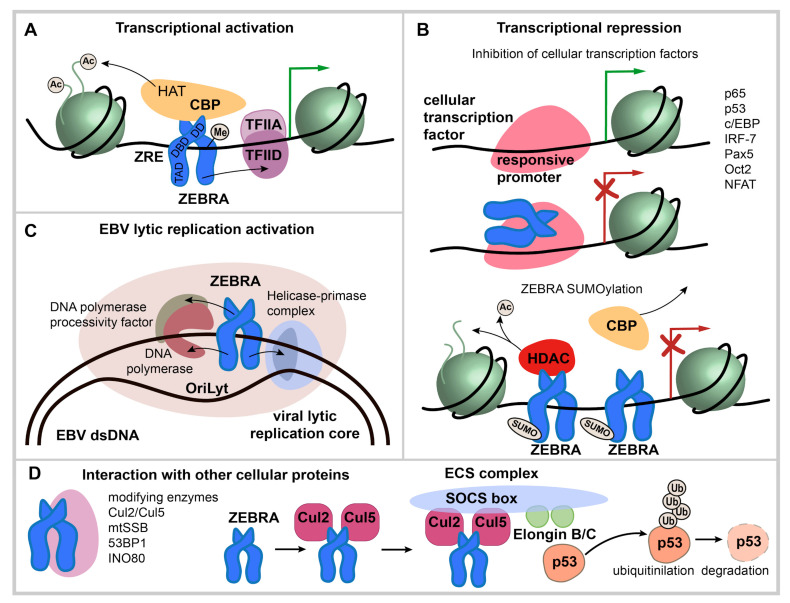
ZEBRA functions. (**A**) transcriptional activation by ZEBRA. ZEBRA is shown as a homodimer, relative positions of transactivation domain (TAD), DNA binding domain (DBD) and dimerization domain (DD) are indicated. ZEBRA binds to specific ZEBRA response elements (ZREs) within promoters of viral and host genes with a preference to methylated-CpG DNA. ZEBRA binding leads to sequential recruitment of basal transcription factors and RNA polymerase II. In addition, ZEBRA binds transcriptional coactivator CREB binding protein (CBP). (**B**) transcriptional repression via ZEBRA binding to cellular transcription factors and by SUMOylated ZEBRA. Transcription factors that interact directly with ZEBRA are listed. The interaction occurs mainly via ZEBRA’s bZIP domain and mutually impedes the function of both ZEBRA and bound transcription factor and results in repression of targeted genes. SUMOylated ZEBRA has a low transactivation activity related to decreased CBP binding and the ability to recruit histone deacetylases (HDAC) to responsive promoters. (**C**) activation of EBV lytic replication. ZEBRA recognizes the EBV lytic origin (oriLyt), serves as the origin binding protein and recruits viral core replication enzymes to initiate lytic replication of EBV. (**D**) interaction with cellular proteins not directly involved in transcriptional regulation. ZEBRA interaction partners are listed. ZEBRA interaction with Cul2/Cul5 induces the formation of multimolecular ECS complex (Elongin B/C-Cul2/5-SOCS-box protein) with the ubiquitin ligase activity that targets p53 for proteasomal degradation.

**Figure 4 cancers-12-01479-f004:**
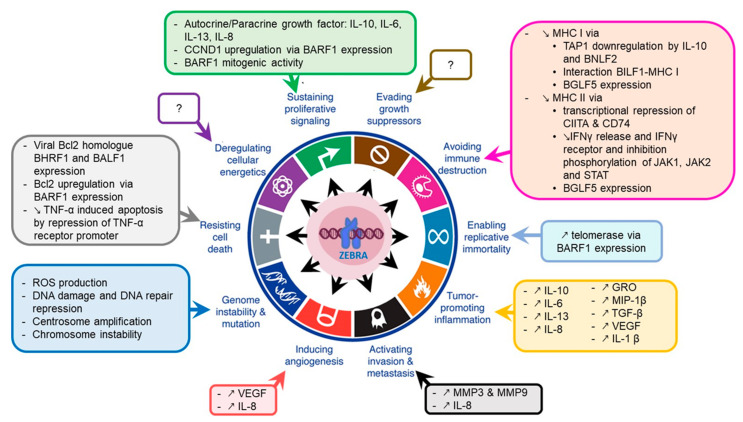
ZEBRA oncogenic properties. ZEBRA directly, or through its target genes, contributes to the acquisition of cancer hallmarks by cells including sustained proliferative signaling, evading or altering the immune response, resisting cell death, enabling replicative immortality, inducing angiogenesis and activating tumor invasion and metastasis. A part of these effects is mediated by genome instability and tumor-promoting inflammation that induce an environment favorable to cancer development and progression. Adapted from [111].

**Table 1 cancers-12-01479-t001:** EBV-associated malignancies in immunocompetent hosts and corresponding EBV association frequency and latent gene expression pattern.

Diseases	% EBV Association	Latency Type ^a^
**B-cells malignancies**		
Burkitt Lymphoma		
• *Endemic BL*	˃95%	I
• *Sporadic BL*	20–80%	I
Classical Hodgkin Lymphoma	20–90%	II
Diffuse Large B Cells Lymphoma (DLBCL)		
• *EBV+ DLBCL Not Otherwise Specified (NOS)*	100%	III
• Pyothorax associated Lymphoma (PAL)	100%	III
**T/NK cells malignancies**		
Extranodal NK/T-cell lymphoma, nasal type	˃95%	I/II
Virus-associated hemophagocytic syndrome T-cell lymphoma	100%	I/II
Angioimmunoblastic T-cell Lymphoma (AITL) ^b^	˃80%	I/II
Hepatosplenic T-cell lymphoma		
Non-hepatosplenic γδ T-cell lymphomas		
Enteropathy-type T-cell Lymphoma		
**Epithelial malignancies**		
Undifferentiated Nasopharyngeal carcinoma	100%	II
Gastric carcinoma	10%	II
Lymphoepithelioma-like carcinoma (salivary gland, tonsils, larynx, thymus, lungs, skin, uterus cervix, bladder, stomach)		
Breast carcinoma		
Hepatocellular carcinoma		
**Mesenchymal malignancies**		
Follicular dendritic cell sarcoma		

^a^ Latency type: Latency I = EBNA1, EBER 1 and 2, BART miRNA; Latency II = EBNA1, LMP1, 2A and 2B, EBER1 and 2, BART miRNA; Latency III = All EBNAs, LMPs, EBERs and BART miRNA. ^b^ In AITL there is no EBV in tumor cells but EBV is nearly always present in tumor B cells, suggesting an indirect role of EBV [6,33]. Blank spaces indicate the missing data.

**Table 2 cancers-12-01479-t002:** EBV-associated malignancies in immunodeficient hosts and corresponding EBV association frequency and latent gene expression pattern.

Diseases	% EBV Association	Latency Type ^a^
**Acquired Immunodeficiency**		
AIDS-associated B cell lymphomas		
• *BL*	30–50%	I
• *Hodgkin Lymphoma*	100%	II
• *DLBCL*		
○ Immunoblastic	70–100%	II/III
○ Non Immunoblastic	10–30%	II/III
○ Central Nervous System lymphoma (CNS)	˃95%	II/III
○ Primary Effusion Lymphoma (PEL)	70–90%	I
○ Plasmablastic lymphoma	60–75%	I
Post-transplantation lymphoproliferative disorder	˃90%	III
Lymphomatoid granulomatosis		
Methotrexate-associated B cell lymphoma		
Leiomyosarcoma		
**Congenital immunodeficiency**		
Severe combined immunodeficiency–associated B cell lymphoma		
Wiskott-Aldrich syndrome–associated B cell lymphomas		
X-linked lymphoproliferative disorder–associated B cell lymphomas		

^a^ Latency type: Latency I = EBNA1, EBER 1 and 2, BART miRNA; Latency II = EBNA1, LMP1, 2A and 2B, EBER1 and 2, BART miRNA; Latency III = All EBNAs, LMPs, EBERs and BART miRNA. Blank spaces indicate the missing data.

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
