# Peer review of "Oncogenic Properties of the EBV ZEBRA Protein"

_cancers, 2020, doi:10.3390/cancers12061479_

Round 1

Reviewer 1 Report

The title is a play on words that can cause confusion and does not indicate, in this case, anything precise or scientifically consolidated.

In the Introduction it is necessary to mention the work of the IARC (A Review of Human Carcinogens. Part B: Biological Agents IARC Monograph on the Evaluation of Carcinogenic Risk to Humans, 2012) to describe the oncogenic role of EBV and its known oncogenic properties.

The work is comprehensively described, but could be better organized. I propose to move section 1.2 EBV-related oncogenesis before section 3.1.

As regards figure 1, there must be better correspondence between the legend and the scheme.

The table is complete and full of information, but it is difficult to consult. It is the opinion of this Reviewer that the table should be divided into three parts: Table 1 should be limited to the so-called "immunocompetent host"; Table 2 should be limited to "Congenital Immunodeficiency"; Table 3 should be limited to "Acquired Immunodeficiency”.

In section 4 ZEBRA in diagnosis and therapy, it might be interesting to discuss the role of ZEBRA, if any, towards the PD-1 / PD-L1 axis

Author Response

Reviewer 1

The title is a play on words that can cause confusion and does not indicate, in this case, anything precise or scientifically consolidated.

The title was changed as follows: “Oncogenic Properties of the EBV ZEBRA Protein”

In the Introduction it is necessary to mention the work of the IARC (A Review of Human Carcinogens. Part B: Biological Agents IARC Monograph on the Evaluation of Carcinogenic Risk to Humans, 2012) to describe the oncogenic role of EBV and its known oncogenic properties.

The work of IARC was acknowledged on page 4 as follows:

“EBV was considered as a class I human carcinogen by WHO (ref: IARC Monograph 1997) and the evidence for EBV-associated oncogenesis have been recently reviewed and updated (ref: IARC Monograph 2012)”.

The work is comprehensively described, but could be better organized. I propose to move section 1.2 EBV-related oncogenesis before section 3.1.

 We do not agree with the reviewer as in section 1.2 we generally introduce the oncogenic properties of EBV and we would like to keep the structure of the review as is.

As regards figure 1, there must be better correspondence between the legend and the scheme.

The figure legend was modified as follows:

(1) Infection occurs after the contact with an infected saliva. (2) After initial infection of oropharyngeal epithelial cells, the virus passes into the underlying lymphoid tissue where it infects the naive B cells. (3) This immediately triggers the transient pre-latent lytic cycle with expression of ZEBRA and other lytic genes involved in resistance to apoptosis and evasion from the immune system. (4) Infected naïve B cells become proliferating B blasts through the growth program (latency III) where all latency proteins are expressed. (5) Cytotoxic T lymphocytes (CTL) trigger a strong immune response (which is impaired during immunodeficiency) to eliminate EBV-infected B cells. (6) Proliferating B blasts will then migrate into the germinal center (GC) and activate the default transcription program (latency II) where latency protein expression is restricted to EBNA1, LMP1 and LMP2. They differentiate into centroblasts and then centrocytes. (7) Centrocytes leave the GC and differentiate into memory B cells circulating in peripheral blood. These cells have turned off the expression of all viral protein (latency 0). (8) Occasionally, circulating EBV-positive memory B cells express EBNA1 during homeostatic cell division to ensure viral genome replication and segregation into daughter cells. (9) Following stimulation, latently infected memory B cells can be recruited into GC. (10) Activated EBV-positive memory B cells can differentiate into plasma cells, reactivate the virus and undergo productive lytic cycle that leads  to (11)  viral shedding into saliva and (12) new naïve B cells infection. (13) Activated EBV-positive memory B cells either reintegrate the pool of memory B cells. But, it is not clear whether  in vivo stimulated EBV-positive memory B cells which have not differentiated into plasma cells undergo an abortive lytic cycle (ZEBRA and early gene expression without viral production) or not, before reintegrating the pool of memory cells. It is also not clear, whether or not, in vivo, these cells successively reexpress the different latency programs in the GC before reintegrating the pool of memory cells.

The table is complete and full of information, but it is difficult to consult. It is the opinion of this Reviewer that the table should be divided into three parts: Table 1 should be limited to the so-called "immunocompetent host"; Table 2 should be limited to "Congenital Immunodeficiency"; Table 3 should be limited to "Acquired Immunodeficiency”. 

We divided the table I into two parts. Now Table I correspond to “EBV-associated malignancies in immunocompetent hosts” and Table II correspond to “EBV-associated malignancies in immunodeficient hosts”. We preferred to keep together the Acquired and Congenital Immunodeficiency as they refer to the same section in the text.

In section 4 ZEBRA in diagnosis and therapy, it might be interesting to discuss the role of ZEBRA, if any, towards the PD-1 / PD-L1 axis

While EBV is related to PD-L1 expression in B cell lymphomas, PD-L1 expression is rather correlated with EBV latency II or III. No significant correlation between PD1 or PD-L1 expression and ZEBRA was found in PTLD and DLBCL cases (PMID:30861172) .

Reviewer 2 Report

Germini and co-authors give a comprehensive review on the EBV immediate early lytic protein ZEBRA.

Most of it appears to be appropriate, but there are some issues to consider:

  1. The title is nice, but only covers a small part of the content. Please consider another title
  2. In the text it would be good to shortly explain the concept of Trojan horse in Greek mythology (I am not sure if it is really relevant to ZEBRA, as it does not appear to be disguised as something that a cell would normally take up)
  3. PLease explain the way the "cell penetrating domain" of ZEBRA works. In addition, I wonder how it can overlap with the DNA binding domain
  4. In figure 1 ZEBRA is not indicated (its gene is). Please consider changing to make it easier to follow (also in some other instances in the text and legends)
  5. Line 115-116:"The mechanism by which EBV participates in the development of human tumors is still unclear and represents a major challenge in cancer research." -> This is a bit too easy... At least add some references

  6. Table I is really not correct. PLease use the most recent WHO classification nomenclature. e.g. EBV+ DLBCL of the elderly does not exist anymore. DLBCL NOS is 0% as this now falls under EBV+ DLBCL. In AITL, hepatosplenic and non-hepatosplenic lymphomas and EATL there is NO EBV in the tumor cells. For nasopharyngeal carcinoma is it only the undifferntiated/non-keratiniziing variant that is Always EBV+, gastric carcinoma is only ~10% EBV+.

  7.   Line 127: growing evidence -> what does this mean? please also include refs.

  8. Quite some abbreviations are not written in full
  9. I don't understand the Rainbow color in figure 2
  10. line 293: It is unclear and and not discussed whether ZEBRA in these tumors is derived from the tumor cells or from EBV ingfected bystander cells. PLease explain or make a reservation (also for other statements where this is unclear, eg line 430)
  11. line 438-439: this is latent infected cells, not lytic, right?

Author Response

Reviewer 2

The title is nice, but only covers a small part of the content. Please consider another title

The title was changed as follows: “Oncogenic Properties of the EBV ZEBRA Protein”

In the text it would be good to shortly explain the concept of Trojan horse in Greek mythology (I am not sure if it is really relevant to ZEBRA, as it does not appear to be disguised as something that a cell would normally take up)

Since we removed Trojan horse from the title as suggested by the Reviewers, we decided not to refer to it in the text either.

Please explain the way the "cell penetrating domain" of ZEBRA works. In addition, I wonder how it can overlap with the DNA binding domain.

The explanation was added to the text on page 10:

CPDs are short sequences with a composition that enables them (and the adjacent protein) to penetrate cells either via endocytotic entry followed by endosomal escape, or by directly penetrating the cell membrane. Their composition is usually either cationic (with a high number of positively charged residues) or amphipathic (with hydrophilic and hydrophobic regions of residues) (PMID: 31443361). ZEBRA CPD is rich in positively charged residues (seven lysines and seven arginines), mostly within DBD (basic region) (Figure S1 A in blue), whereas hydrophobic amino acids (one valine, five alanines, seven leucines) of CPD are mostly within DD (leucine zipper) (Figure S1 B in red) (PMID: 20385549). In another protein possessing the CPD, HIV-1 Tat, the CPD  region is also multifunctional (PMID: 30609200). Presumably, the cationic part serves for interaction with the negatively charged phosphate groups of membrane phospholipids as well as on DNA, while hydrophobic residues interact with the hydrophobic part of the phospholipid membrane and participate in ZEBRA’s dimerisation (Figure S1 C).

Figure S1. ZEBRA’s cell penetration domain. A, positions of positively (blue) and negatively (red) charged residues in ZEBRA’s structure. Residues from 175 (top) to 236 (bottom) were characterized by X-ray crystallography (PMID: 16483937) and are available at SWISS-MODEL Repository (PMID: 14681401). B, the same ZEBRA structure, positions of hydrophilic (blue) and hydrophobic (red) residues are shown. C, primary structure of ZEBRA’s cell penetration domain (residues 170-220). Positively charged amino acids (seven lysines and seven arginines) are shown in blue, whereas hydrophobic amino acids (seven leucines, five alanines, one valine) are shown in red (PMID: 20385549).

In figure 1 ZEBRA is not indicated (its gene is). Please consider changing to make it easier to follow (also in some other instances in the text and legends)

We highlighted Zebra in Figure 1 and in the legend to make it easier to follow as suggested by the reviewer.

Line 115-116:"The mechanism by which EBV participates in the development of human tumors is still unclear and represents a major challenge in cancer research." -> This is a bit too easy... At least add some references

We have modified the phrase on page 7 as follows:

“Many mechanisms of EBV related oncogenesis have been proposed and a possible role for different EBV components has been described (reviewed in [7,27,29–32]). Nevertheless, even if great progress has been made in understanding the EBV links to cancers, many aspects of EBV- related oncogenesis are still unknown and represent a major challenge in cancer research.”

Table I is really not correct. PLease use the most recent WHO classification nomenclature. e.g. EBV+ DLBCL of the elderly does not exist anymore. DLBCL NOS is 0% as this now falls under EBV+ DLBCL. In AITL, hepatosplenic and non-hepatosplenic lymphomas and EATL there is NO EBV in the tumor cells. For nasopharyngeal carcinoma it is only the undifferntiated/non-keratiniziing variant that is Always EBV+, gastric carcinoma is only ~10% EBV+.

We agree with the reviewer and we have changed the Table 1 accordingly:

  • The DLBCL classification has been corrected according to the 2016 WHO classification nomenclature.
  • Indeed, in AITL there is no EBV in tumor cells but EBV is nearly always present in B cells from tumor biopsies, suggesting an indirect role of EBV (IARC Monograph 2012, PMID: 17555446). We add a commentary after the table to clarify that in this case EBV is present in B-cells.
  • EBV positivity in neoplastic cells has been described in some cases of hepatosplenic T-cell lymphoma (PMID: 10672057) even though we could not find data related to the frequency of EBV association for these lymphomas.
  • We precised that only undifferentiated NPC are always associated with EBV.
  • Indeed, only 10% of gastric carcinoma are associated with EBV. It was a typing error on our part.

Line 127: growing evidence -> what does this mean? please also include refs.

We have clarified this point modifying the phrase on page 8 as follows:

“Even though the lytic cycle was long assumed to inhibit tumorigenesis due to final lysis of the infected cells, an increasing amount of data support its contribution to oncogenesis mainly at its initiation or through the abortive lytic cycle and/or autocrine or paracrine effects of EBV IE proteins [27,33,34].”

Quite some abbreviations are not written in full

All the abbreviations in the text and the abbreviation list has been checked and corrected where required

I don't understand the Rainbow color in figure 2

ZEBRA’s full tertiary (or quaternary) structure wasn’t resolved. Only three parts of the structure were characterized by crystallography. Three available partial 3D structures were imported from the  SWISS-MODEL Repository (PMID: 14681401) (accession number P03206) and are based on crystal structure data published by (PMID: 16483937, PMID: 22343629, PMID: 16275762). They are shown below the respective primary sequence. Rainbow color code is used to map approximate residue position concordance between primary and tertiary (or quaternary) structure.

We changed figure legend accordingly:

Three available partial 3D structures were imported from SWISS-MODEL Repository (PMID: 14681401) (accession number P03206) and are based on crystal structure data published by (PMID: 16483937, PMID: 22343629, PMID: 16275762). They are shown below the respective primary sequence. Rainbow color code is used to map approximate residue position concordance between primary and tertiary (or quaternary) structure.

Line 293: It is unclear whether ZEBRA in these tumors is derived from the tumor cells or from EBV infected bystander cells. PLease explain or make a reservation (also for other statements where this is unclear, eg line 430)

ZEBRA protein of RNA was found in tumor cells or tumor biopsies. We have added this information to page 16:

“The presence of ZEBRA protein or its mRNA was also reported in tumor cells or in tumor tissue biopsies in other types of EBV-induced lymphomas, such as HL, DLBCL and BL [85–88].”

Line 438-439: this is latent infected cells, not lytic, right?

We indeed meant latent infected cells. We have corrected the text.